# Patterns and drivers of water quality changes associated with dams in the Tropical Andes

R. Scott Winton[1,2,3], Silvia López-Casas[4,5], Daniel Valencia-Rodríguez[5,6,7], Camilo Bernal-Forero[8], Juliana Delgado[9], Bernhard Wehrli[1,2], Luz Jiménez-Segura[5]

[1]Institute of Biogeochemistry and Pollutant Dynamics, ETH Zurich, 8092 Zurich, Switzerland
[2]Department of Surface Waters, Eawag, Swiss Federal Institution of Aquatic Science and Technology, 6047 Kastanienbaum, Switzerland
[3]Stanford Doerr School of Sustainability, Stanford University, California, USA
[4]Wildlife Conservation Society Colombia, Bogotá
[5]Grupo de Ictiología, Instituto de Biología, Universidad de Antioquia, Medellín, Colombia
[6]Fundacion Horizonte Verde, Cumaral, Colombia
[7]Red de Biología Evolutiva, Instituto de Ecología AC, Xalapa 91070, Veracruz, México
[8]Autoridad Nacional de Licencias Ambientales, Bogotá, Colombia
[9]The Nature Conservancy Colombia, Bogotá

*Correspondence to*: R. Scott Winton (scott.winton@gmail.com)

**Abstract.** The Tropical Andes is a biodiversity hotspot facing pressure from planned and ongoing hydropower development. However, the effects of dams on river ecosystems of the region as mediated by physico-chemical changes to water quality are poorly known. Colombia is unique among its peers in South America for managing central public environmental databases, including surface water quality data sets associated with environmental monitoring of dams. To assess the relationship between hydropower and Colombian river conditions, we analyze monitoring data associated with 15 dams, focusing on oxygen availability, thermal regimes and sediment losses because these properties are influenced directly by river damming and impose fundamental constraints on the structure of downstream aquatic ecosystems. We find that most Colombian dams (7 of 10) seasonally reduce concentrations of total suspended solids by large percentages (50-99%) through sediment trapping. Most dams (8 of 15) also, via discharge of warm reservoir surface waters, seasonally increase river temperatures by 2 to 4°C with respect to upstream conditions. A subset of four dams generates downstream hypoxia (<4 mg L$^{-1}$) and water 2 to 5°C colder than inflows—both processes driven by the turbination and discharge of cold and anoxic hypolimnetic waters during periods of reservoir stratification. Reliance on monitoring data likely leads us to under-detect impacts because many rivers are only sampled once or twice per year and cannot capture temporal shifts across seasons and days (i.e., in response to hydropeaking). Despite these blind spots, the monitoring data point to some opportunities for planners and hydropower companies to mitigate downstream ecological impacts. These findings highlight the importance of implementing environmental monitoring schemes associated with hydrologic infrastructure in developing countries.

## 1 Introduction

Freshwater ecosystems and the services they provide to society are threatened worldwide, especially biodiversity (He et al., 2019) and fisheries resources (Deines et al., 2013; Stone, 2016). One of the major threats is the construction of dams for hydropower generation, which is booming especially in tropical regions (Zarfl et al., 2014) where biodiversity is high (Ailly et al., 2014), as is fish protein dependence for poor rural populations (Kirby et al., 2010). Hydropower development is routinely promoted for its potential to meet United Nations Sustainable Development Goals (SDGs), but negative side

effects of dam construction may create obstacles for achieving SDGs.

The benefits of hydropower are well understood—namely the provision of a renewable source of energy with a potentially low carbon emissions intensity (Almeida et al., 2019), thereby contributing toward SDG 7 (affordable and clean energy) and SDG 13 (climate action). Critics, however, argue that the external costs of hydropower are frequently overlooked (Opperman et al., 2015). Dams cause the fragmentation of rivers (Anderson et al., 2018), disrupt hydrologic regimes critical for life

histories of potamodromous freshwater fish (de Fex-wolf et al., 2019; Carvajal-Quintero et al., 2017), and alter the downstream physico-chemical condition of river water (Winton et al., 2019).  These threats to freshwater ecosystems misalign with SDG 14 (life below water), SDG 2 (zero hunger) and SDG 1 (no poverty).

Although retrospective case studies have been able to identify major ecological changes associated with dam construction, catastrophic ecological consequences of damming are not inevitable (Winemiller et al., 2016). Careful siting, design and

operation can potentially mitigate the most costly of externalities associated with new reservoirs for hydropower (Moran et al., 2018; Opperman et al., 2017). With thousands of new projects under construction and in a planning phase globally (Opperman et al., 2015, 2017; Zarfl et al., 2014), the amount of work needed to analyze each case easily outpaces the available expertise, especially since hydropower planning and impact assessment spans financial, economic, social, engineering, hydrologic and ecologic dimensions. Synthetic assessments of existing hydropower portfolios have the potential

to identify guidelines that could provide a shortcut for planners, helping them prioritize projects less likely to cause ecological harm and shelve more problematic proposals. Such assessments have been conducted at global scales with a focus on specific processes, such as sediment transport (Maavara et al., 2015; Vörösmarty et al., 2003), fragmentation (Grill et al., 2015), greenhouse gas emissions (Harrison et al., 2021), water supply (Opperman et al., 2017), and water quality (Winton et al., 2019).

Global-scale analyses, however, may have limited applicability to local-scale contextual realities and be less useful for decision-makers and on-the-ground managers. Regional studies may provide an important role of providing synthetic understanding from a portfolio of existing hydropower projects while remaining relevant for a specific geography (Flecker et al., 2022; Kummu et al., 2010). In this study, we aim to provide a gap-bridging regional assessment focused on one of the many regional concentrations of hydropower expansion, the Tropical Andes of South America, a hotspot for freshwater

biodiversity with 967 known endemic species and 17.5% of fish species at risk of extinction (Tognelli et al., 2016). The region is rich in untapped hydropower resources thanks to steep elevational gradients and humid climates. Among the

countries representing the region (including Peru, Ecuador and Bolivia), Colombia stands out as an ideal case for study because it maintains a centralized and publicly available data repository for all hydropower monitoring data and hosts roughly 40% of the region's hydropower dams (World Register of Dams, 2018) (Fig. 1; Table 1).

We limit our focus to alterations of physico-chemical parameters–temperature, dissolved oxygen and total suspended solids. These parameters are included in most aquatic environmental monitoring programs and define the critical oxic living conditions for aquatic fauna and the dynamic reconstruction of riparian habitat. Thermal shocks or regime changes are known to disrupt the life cycles of tropical biota (Caissie, 2006; Ward and Stanford, 1982), with cold water discharges from dams frequently cited as negatively impacting fish (King et al., 1998; Todd et al., 2005; Preece and Jones, 2002; van Vliet et

al., 2013; Cooper et al., 2019). Changes to the availability of dissolved oxygen impose physiological constraints on the potential metabolic activity of aquatic animals (Ekau et al., 2010), with all but the most adapted of macroscopic fauna unable to tolerate hypoxia (Agostinho et al., 2021; Chapman et al., 2002; Kramer and McClure, 1982). Disruptions to sediment transport rob floodplains and deltas of a critical lifeline (Giosan et al., 2014; Constantine et al., 2014) and can even restructure fish communities (Granzotti et al., 2018).

Our goal is to systemically assess Colombian dams for impacts on temperature, dissolved oxygen and total suspended sediments through analysis of available monitoring data. Since it is the mixing behavior of reservoirs that largely governs thermal and oxygen dynamics surrounding dams, we assess the Colombian dam portfolio for evidence of thermal stratification. For the parameters themselves, we compare downstream monitoring stations to those upstream, which serve as a reference condition for the state of the river. We ask: do Colombian reservoirs stratify? Does the stratification lead to

alteration of river thermal regimes? Does stratification lead to hypoxia in downstream waters? How effectively do dams trap sediment and reduce the concentration of suspended matter? The answers to these questions have important implications for planning the future of hydropower development in Colombia and throughout the Tropical Andes biodiversity hotspot. We assess which characteristics of dams appear to be associated with the most problematic outcomes and provide recommendations to decision-makers and regulators for how to minimize harm moving forward.

**2 Methods**

**2.1 Colombian Dams and monitoring data**

Colombia's Autoridad Nacional de Licencias Ambientales (ANLA) maintains environmental monitoring databases for all of Colombia's hydropower projects generating >100 MW (26 power stations). Hydropower installations are distributed throughout the country's three branches of the Andes, but concentrated in the Magdalena-Cauca basin, which is home to

95 roughly 70% of the total national population of 50 million people. Colombian hydropower installations span a wide range of elevations—from 70 m in the foothills to more than 2200 m in the high Andes, with a correspondingly high range of mean ambient temperatures. Because they are distributed across the many slopes of the Colombian Andes including within relatively dry inter-Andean valleys, they experience highly divergent precipitation regimes, including monomodal and

bimodal rainfall patterns. The west slope of the western Andes is one of the Earth's wettest regions, with some
municipalities reporting annual rainfall of more than 10,000 mm $y^{-1}$, in contrast in the Sogamoso reservoir near
Bucaramanaga receives roughly one-tenth this much annual rainfall. The climate settings of Colombia dams are enormously
diverse. Of the hydropower projects under ANLA jurisdiction, 22 are associated with a dam and reservoir. The remaining 4
are micro-hydro systems that divert a portion of river discharge into turbine intakes, but do not completely impound their
associated rivers, so we exclude them from analysis.

As part of the environmental licensing process, the companies that operate the hydropower projects are required by ANLA to
monitor the environmental impacts of their operations and submit annual reports and monitoring data sets. The companies
(which may be public or private) contract environmental consulting firms to collect field samples and analyze them in their
laboratories. The data are assimilated into a central georeferenced database maintained by ANLA, which screens the data for
quality. In a legal sense, these data are public and ANLA works to guarantee access, but as of this work, plans to build an
online portal to facilitate direct data acquisition have not yet been implemented. Currently data is available to the public via
direct request to ANLA. For this study, we requested and were granted access to systematized hydropower monitoring data
from years 2017 and 2018, which were the most recent years of fully quality controlled data (Table 1). ANLA is working to
incorporate historical data into its database, but since our primary goal was to examine changes across the portfolio of
hydropower systems rather than examine evolution of dammed rivers over time, we did not request data from previous years.

Data are stored in .gdb files, which require graphical information systems software (i.e. ESRI ARCMap) to read. We
screened all data for statistical outliers and for chemical/physical plausibility and did not find the need to discard any data
from our focal parameters.

Data collection frequency (summarized in Table 1) is highly heterogeneous across sites, with two dams (Urra and Porce 3)
getting monthly measurements, while the majority have just one to three data points per year. Sampling frequency is
generally higher for more recently constructed reservoirs, reflecting an update to regulations in 1993. Older reservoirs (pre-
2000) are generally monitored only once or twice per year. Each dam's monitoring approach is uniquely tailored to its
geographic circumstances, but generally includes surface water sampling from one or more stations upstream of the reservoir
and downstream of the dam as well as samples from various parts of the reservoir itself, often including depth profiles.
Adding to the data heterogeneity is a lack of standardization leaving key fields missing in some entries. For example,
Colombia's largest dam by volume, Hidrosogamoso Reservoir, has been extensively profiled, but the depth and time
information are omitted, limiting interpretability.

## 2.2 Analytical approach

Since stratification is a fundamental driver of thermal and oxygen dynamics within and downstream of hydropower
reservoirs (Winton et al., 2019), we first qualitatively assessed depth profiles (where available) for stratification strength.
Since the profile data availability is quite variable, including the number of depth points sampled and seasonal coverage, we
focus on deep water oxygen levels below the thermocline and the magnitude of the $O_2$ concentration difference between

surface and deep waters to sort each reservoir into coarse categories of strongly-, weakly- or non-stratifying. We classified reservoirs as strongly-stratifying if they showed $O_2$ concentrations of <2 mg $L^{-1}$ at depth and a difference between the surface of at least 3 mg $O_2$ $L^{-1}$. All other reservoirs we classified as weakly-stratifying as all showed differences in $O_2$ and temperature between surface and deep waters of at least 2°C and 1 mg $O_2$ $L^{-1}$ (for summary, see Table S1). Secondarily we use the densiometric Froude number (Parker et al., 1975), which compares the inertial force of reservoir water, based on mean flow-through velocity with the gravitational force tending to maintain densiometric stability (Orlob 1983, Deas 2000). The Froude number can be approximated using a simplified formula with reservoir length (L), depth (D), discharge (Q) and volume (V): F = 320(L/D)(Q/V) (Parker et al., 1975), as applied as a metric for stratification behavior (Ledec and Quintero, 2003).

To understand dam impacts, we rely on contemporaneous paired measurements of upstream versus downstream conditions, and interpret differences to be attributable to dam effects, a widely used approach (Fovet et al., 2020). Colombian regulations stipulate that industrial or commercial activities should not alter water temperatures by more than 5°C, a temperature difference associated with acute responses from tropical biota, such as fish mortality from warm water (Cooper et al., 2019) or disruption to fish reproductive cycles from cold water (King et al., 1998). A more conservative thermal change threshold of ±2°C may be warranted, given that community composition of macroinvertebrates is highly sensitive to subtle changes to thermal regime (Eady et al., 2013; Preece and Jones, 2002) and assessments of global climate change effects on fish delimit a 2°C threshold for impacts (van Vliet et al., 2013). Furthermore, since we are limited to just a few random comparisons for most sites by data availability for Colombia, we are unlikely to be capturing the most extreme moments of thermal impact and so the precautionary principle would dictate a stricter approach. We therefore consider a change of ±2°C to be evidence for warmwater or coldwater pollution, respectively. For changes in dissolved oxygen, we found that a loss of >2 mg $L^{-1}$ imposed by dams always corresponded to a downstream concentration of <5 mg $L^{-1}$, which is the regulatory minimum concentration for cold freshwaters in Colombia. Oxygen availability imposes a fundamental constraint on many aquatic species (Coble, 1982; Spoor, 1990; Ekau et al., 2010) and we therefore assess impact along a change threshold of 2 mg $L^{-1}$ to distinguish between "minor" and "severe" oxygen loss. For sediment trapping, although the literature is clear about the potential consequences of dam-induced sediment loss from rivers, choosing an appropriate threshold demarcating what constitutes a severe loss is highly subjective and there are no regulatory guidelines for total suspended solids. We report the gradient of responses, noting how many dams trapped >50% and >99% of inbound suspended sediments.

We note that in some cases dams are oriented in a cascade whereby the outflow from one reservoir rapidly (or immediately) enters the reservoir for the next power station. In such a configuration the inflowing water does not necessarily represent a neutral reference condition as it is likely to already be altered by the previous dam. This may bias us to underestimate the potential of lower chain dams to alter water quality, but without access to data on pre-dam river conditions, there is no alternative metric for reference conditions beyond upstream waters. Since only one year of recent data is available for most projects, we focus our analyses on just the most recent year (2017 or 2018) under the assumption that covering an annual climate cycle is more important to answering our questions than studying variation between years.

# 3 Results

## 3.1 Stratification

Our analyses of reservoir mixing behavior indicate that most, if not all, Colombian reservoirs stratify strongly. Of the 22 reservoirs evaluated, 12 had available depth profile information, and of these, eight exhibit anoxia (DO < 1.5 mg/L) in deep waters, indicating that they stratify sufficiently to prevent consistent reoxygenation from the surface (Table S1). The remaining four reservoirs lack evidence for acutely hypoxic deep-water, however, with just two or three depth profiles per year, it is possible that this limited sampling did not coincide with periods of stronger stratification, which can develop rapidly in tropical lakes (Lewis, 1996). From the few reservoirs that have been profiled several times per year, it appears that Colombian reservoirs are rarely, if ever, well-mixed (Figs. S1, S2, S3). This result is supported by our calculations of densiometric Froude Number (Fr)(Parker et al., 1975; Orlob, 1983), which indicate that, of the 12 dams with discharge data available (necessary for calculating Fr), all except for Guatape, fall into the strongly stratifying domain of Fr < 0.3 regardless of whether mean depth or maximum depth is used for the calculation (Fig. 2). In Colombia, large reservoirs tend to stratify strongly, and this limnological reality creates the potential for thermal and biogeochemical disruptions to downstream aquatic ecosystems.

## 3.2 Thermal regime change

Consistent with the expectation that stratifying reservoirs will alter thermal regimes, we find that nine of the 12 (75%) Colombian reservoirs assessed generated temperature anomalies in the river of at least 2˚C (Fig. 3). Four reservoirs create cold-water pollution of at least -2 ˚C downstream and seven create warm-water pollution of at least +2 C downstream. Two reservoirs—Urrá and Quimbo—generated both cold- and warm-water pollution at different times of the year, illustrating the importance of temporal dynamics. Seasonal climate cycles that govern stratification, fluctuations in inflowing discharge associated with droughts/floods and dam operation (i.e. hydropeaking) may all influence the direction and magnitude of downstream thermal effects over timescales of months to minutes. Our ability to assess more broadly thermal changes is severely limited by the lack of frequent (seasonal or monthly) and detailed (with depth profiles) monitoring for most projects.

## 3.3 Hypoxia

We find that minor loss of oxygen (dDO < 2 mg/L) in waters downstream of Colombian dams was common (evident in nine of 15 reservoirs), but severe oxygen loss relative to upstream (dDO > 2 mg/L) was associated with only four reservoirs (Fig. 4). These hydropower schemes (Urrá, Quimbo, Sogamoso, Prado) are the same that exhibited cold-water pollution (Fig 3), a coincidence that supports our assumption that both effects are driven by thermal stratification (Hutchinson and Loffler, 1956; Lewis, 1987). As with temperature, we observe high seasonal variation in upstream-downstream oxygen dynamics, which, again, supports the notion that hypoxic effects are sensitive to the seasonality of climate, stratification, and dam operations.

### 3.4 Sediment trapping

Loss of total suspended sediments (TSS) associated with Colombian dams is pervasive and for several reservoirs, extreme (Fig. 5). Six out of the 10 reservoirs we assessed, showed decreases of TSS of more than 50% and two dams—Quimbo and Sogamoso—logged losses of more than 99%. Some reservoirs exhibited an increase in TSS downstream of the dam relative to upstream, which may reflect local erosion processes or activities, while other reservoirs exhibit upstream sediment losses. For example, the Porce river has been dammed by several dams in a cascade system and as a result carries a relatively low sediment load, however, the river reach immediately below Porce 3 dam is turbid because of local illegal mining activity. Beyond such local artifacts, turbid inflow and clear outflow is a typical modality of Colombian dams, especially during rainy periods.

## 4 Discussion

### 4.1 Drivers

In contrast to sediment loss and warm-water discharge, which appear to be ubiquitous, cold-water pollution and hypoxic effects appear to both afflict the same subset of four Colombian reservoirs (Urrá, Quimbo, Prado, Sogamoso). Examining the characteristics of these reservoirs reveals some important commonalities. All are very deep (at least 70 m) and have long mean residence times, sufficient for anoxia to develop in the hypolimnion (as is evident at Urrá, which has depth profile data; Fig. S1). Since all the powerplants have fixed depth intakes to run their turbines, there is no opportunity for operators to avoid discharging cold and hypoxic water when the thermocline/oxycline is shallower than the depth of intakes. In the case of Quimbo, the hydropower authorities are injecting liquid oxygen into discharge waters to meet the minimum dissolved oxygen requirement of 4 mg $L^{-1}$, an expensive measure they hope will be phased out by 2023 as oxygen demand in the reservoir lessens, as typically happens as reservoirs age. For Urrá, the authorities found a policy solution, passing a rule relaxing the minimum oxygen requirement to 2 mg $L^{-1}$ for the first 5 km downstream of the dam, which is good for hydroelectric generators, but does not reduce impacts on aquatic biodiversity and its associated ecosystems services. Both Prado and Sogamoso are out of compliance with the 4 mg $L^{-1}$ threshold and the authorities are evaluating management options (unpublished documents from Autoridad Nacional de Licencias Ambientales).

The development of hypoxia and lakes and reservoirs can be exacerbated by long hydraulic residence times and elevated oxygen demand from organic matter inputs. Colombian dam data illustrate the importance of these reservoir and catchment characteristics for determining risks of hypoxia developing in downstream rivers. La Miel in some important respects appears to be like Quimbo—they have similar depths, hydrologic residence times and have fixed-depth intakes—but in contrast to Quimbo, La Miel shows no sign of downstream hypoxia. La Miel also relies on a reoxygenation system—an air bubbler in its oscillation cavern—but this is a much less intensive and less costly intervention than Quimbo's need for liquid oxygen injection. Part of La Miel's behavior may be attributable to the fact that its inflowing water is near-saturation and has

very low biochemical oxygen demand (BOD; Table S2) meaning that severe hypoxia only appears rarely and typically well-below 50 m depth (Fig. S2). This is likely a function of the well-preserved state of the La Miel catchment, which includes

robust riparian buffers that buffer the reservoir against extreme hydrologic fluctuations. In contrast, Porce 2, which lies downstream of the Medellin metropolis with a population of 4 million, receives inflows that are already hypoxic ($<5$ mg L$^{-1}$) and have elevated BOD (Table S2). Therefore, anoxia downstream of Porce 2 is not surprising.  Further downstream, the Porce 3 reservoir receives river water with substantial oxygen deficits, but its short residence time of just over eight days prevents the dam from discharging water with significantly less oxygen than its inflows. Although the evidence from

Colombian dams is anecdotal, it is consistent with the conventional logic that reservoirs loaded with high levels of organic matter from inflows (or left behind by inundated terrestrial ecosystems during reservoir filling), and with long residence times, have greater potential to develop hypolimnetic anoxia for discharge downstream.

Dam design features, rather than environmental factors such as oxygen demand, offer an alternative explanation for why some dams avoid discharging hypoxic water downstream. A tower system with multiple intakes spanning 46 vertical meters

in the Chivor reservoir allows for discharged waters to be sourced from different depths as water levels fluctuate seasonally. Although we see no evidence for hypolimnetic hypoxia in this reservoir (Table S2), its selective withdrawal system could ensure oxic surface waters are passed downstream, as has been proposed as a solution to tailwater hypoxia in other tropical dams (Kunz et al., 2013). At La Miel dam an air blower in the oscillation cavern provides reaeration of seasonally hypoxic (2 to 5 mg L$^{-1}$) turbinated waters, such that discharged waters typically maintain DO concentrations of at least 6 mg L$^{-1}$ (Table

S2). Low biological oxygen demand of inflows, as described above, may be an important mitigating factor, but the evidence for the effectiveness of an engineered reoxygenation system for avoiding downstream hypoxia cannot be ignored.

## 4.2 Implications for river ecology

Many tropical freshwater aquatic species are highly susceptible to thermal regime changes (Olden and Naiman, 2010) and a temperature change of a few degrees (Fig. 3) may very well cause disruptions to the life cycles of sensitive species (King et

al., 1998; Clarkson and Childs, 2000). In the Andes, surface-releasing reservoirs that discharge warm waters might hypothetically favor lowland species over cooler water species that we would normally expect to be present at a given altitude. Several potamodromous fish in Colombia's Magdalena river, such as Bocachico (*Prochilodus magdalenae*) and Pimelodids (*Pseudoplatystoma magdaleniatum*, *Pimelodus yuma*) migrate seasonally for spawning from lowlands to up to 1200 m or 500 m elevation, respectively, historically transiting river reaches that have been dammed in recent decades

(López-casas et al., 2014; López-Casas et al., 2016). Therefore, these species may avoid spawning in rivers altered by upstream dams, effectively reducing available reproductive habitat (López-Casas, 2015). Alternatively, if the fish do spawn, thermal changes may disrupt the timing of embryo development, and can even be lethal as it has been documented in Colombian fish farms (Harvey and Hoar, 1980) and the Mekong River of tropical Southeast Asia (Li et al., 2021). Researchers have found that hydropower generation is associated with changes in production of a hormone driving oocyte

maturation of the *P. magdalenae,* thus disrupting its spawning cycle in dammed rivers (de Fex-wolf et al., 2019). For

Andean fish species, ranges of thermal tolerance are poorly known, so more research would be needed to test such a hypothesis.

Hypoxic conditions in the tailwaters of dams impose even more dramatic ecological constraints than temperature. Dissolved oxygen concentrations below 3.5 to 5 mg L$^{-1}$ trigger escape behavior in most macroscopic organisms (Spoor, 1990) and monitoring data reveals concentrations of less than 5 mg L$^{-1}$ below seven Colombia hydropower dams. Because the data comes from sparse grab samples, it is not clear how persistent hypoxia is in these river reaches, but at minimum, they indicate a loss of viable habitat for hypoxia-sensitive species for at least some parts of the year. The fish communities downstream of Colombia's Porce 3 dam—one of the projects with hypoxic tailwater—have shifted, with a loss of some native species and replacement by invasive exotics (Valencia-Rodríguez et al., 2022). The authors of this study attribute these shifts to habitat fragmentation, but anoxia/hypoxia could be an important factor as many other studies have commented that the multiple changes and stressors imposed by dams are difficult to disentangle (van Puijenbroek et al., 2021; Young et al., 1976). A review from Brazil indicates that poor oxygen management associated with dams is likely to be a major driver of fish mortality in South American rivers, accounting for roughly 40% fish kills covered by media (Agostinho et al., 2021). As with temperature, a limited of understanding of species-specific fish tolerance thresholds makes it impossible to fully understand the impact of hypoxia/anoxia on fish communities.

Sediment trapping in dams, reaching 99% efficiency in some Colombian cases, exerts ecological impacts at different spatial scales. Unnaturally clear river water below dams may favor a different pool of top predators, those adapted for visual hunting, which may displace species adapted to turbid conditions, setting off trophic cascades rippling down the food chain. This effect has been documented at a large dam in Brazil (Granzotti et al., 2018), but not to-date in the Tropical Andes. If sediment load is not replenished through additional erosion in excess of deposition downstream of the dam, then ultimately floodplain and delta ecosystems, which depend on riverine sediment delivery, will be starved, disrupting cycles of nutrient retention and transport (Kondolf et al., 2014). A modelling exercise estimates that in the heavily dammed Magdalena River Basin up to 40% of sediments are currently being trapped behind dams and this figure could increase to up to 68%, threatening the ecological functioning of the Colombia's Mompós depression, one of South America's largest wetland complexes (Angarita et al., 2018). Scientists have sounded the alarm that, globally, sediment trapping at dams in concert with sea level rise may lead to a massive loss of coastal deltaic wetlands (Giosan et al., 2014; Dunn et al., 2019). Colombian river sediments support mangroves on the Pacific and Caribbean coasts, though the extent to which dams may be impacting floodplain lakes, coastal mangroves and deltaic processes within Colombia's coastal zones is unclear.

It is a challenge to pinpoint the exact mechanism by which dams alter their associated aquatic ecosystems because they impose so many changes, spanning physical, hydrological, chemical and biological dimensions, simultaneously (Young et al., 1976). Many studies of dammed river ecology focus on hydromorphological changes in habitat structure or availability, loss of connectivity and alterations to flow regimes (Bratrich et al., 2004; García et al., 2011). In this study, we focus on oxygen, temperature and suspended sediments and find that these are just as plausible mechanistic pathways that reduce habitat quality and availability, therefore, driving shifts in ecological communities in dam-adjacent ecosystems.

Environmental assessments of dams should take care not to ignore the changes dams impose on the physico-chemical condition of downstream waters and underestimate temperature, oxygen and suspended sediments as modes of ecological change.

### 4.3 Implications for management

### 4.3.1 Regulators

The frequency of problematic changes to river temperature, oxygen and sediment associated with dams suggests that regulators should consider risk-assessments specific to these parameters and their associated ecological side-effects in environmental impact assessments of new dam projects. It may even be worth assessing some older existing dams for which impacts have been underappreciated because of sparse monitoring requirements. Our analyses of monitoring data show that measurements frequent enough to capture seasonal variability are highly valuable. Just six Colombian reservoirs had three or

more sampling dates per year, making it difficult to assess the effects of seasonality on physico-chemical properties for most Colombian hydropower projects. For reservoirs with few measurements, we may be missing important seasons and it is probable that hypoxia downstream of Colombian dams is more prevalent than what happens to be captured by these sparse observational "snap shots". From sites with more frequent measurements, it is possible to glean a deeper understanding of the interactions between reservoir limnology and downstream conditions relative to upstream reference. For example, the

monthly monitoring scheme at Urrá reservoir allowed us to observe that downstream oxygen concentrations were always depleted relative to upstream even though the reservoir hypolimnion was only hypoxic for part of the year (Fig. 6A). Monthly monitoring at Urrá also gives a much more complete picture of thermal outcomes. It is evident that the Urrá reservoir remains thermally stratified 12 months out of the year and that while upstream river temperatures fluctuated some 6˚C seasonally, downstream temperatures are more homogenous varying less than 3˚C throughout the year (Fig 6B). For 8 of

12 months of the year, Urrá exerted only minor (< 2 ˚C) thermal effects on downstream waters. If we were to randomly select three months of data for Urrá, there would be a 42% chance of not detecting one of the more extreme outcomes. For reservoirs with one (e.g., Betania) or a few measurements per year (e.g. Porce 2), we are undoubtedly missing most of the range of their potential thermal impacts. It is likely that downstream thermal regime changes are even more prevalent than our analysis of available data seems to reveal. Because Urrá also employs many monitoring stations extending downstream

of the dam, reoxygenation and warming can be tracked longitudinally as the river flows. This makes it possible to assess, for example, the length of river reach for which dissolved oxygen is below a certain target threshold. It would be much easier to detect and understand impacts if more hydropower dams had such high frequency and spatially rich monitoring programs.

   Of course, increasing monitoring effort will increase costs, so regulators will have to decide whether the utility of higher frequency and higher quality information justifies the expense. In addition, monitoring is not practical for detecting

fluctuation in water conditions that occur over timescales of weeks, days or hours. Automated sensors placed strategically below reservoirs may be a cost-effective method for generating high-frequency measurements of dissolved oxygen,

temperature, turbidity and other parameters. Although sensors cannot replace manual grab samples and laboratory analyses, which cover a much broader suite of parameters, they do provide a window into sub-daily fluctuations in water properties and may be effective for some classes of solutes (Pesántez et al., 2021). Hourly measurements can give insights into how hydropower operations (i.e. hydropeaking) interact with water conditions (Calamita et al., 2021), or capture the effects of episodic events such as a reservoir drawdown for maintenance. Monitoring remotely via satellite is another potential low-cost monitoring solution. Remote sensing would not work for dissolved oxygen but may be viable for tracking changes in temperature and chlorophyll-a and has been effectively applied toward monitoring big shifts in river turbidity (Rudorff et al., 2018). We caution that satellite data alone, even in the most ideal of scenarios, still requires ground-truthing, so cannot completely replace ground-based monitoring.

We note that our analyses are only possible because Colombia has a centralized and public repository for environmental monitoring data, a system uncommon in the tropics and which puts it ahead of its peers. Neighboring countries along the Tropical Andes—Venezuela, Ecuador, Peru, Bolivia—all lack a national-scale public data repository for hydropower monitoring data, making systematic assessment of national portfolios difficult. Colombia's database, mandated and curated by ANLA, probably provides the best opportunity to understand river responses to hydropower for the Tropical Andes region.

### 4.3.2 Planners

For siting and designing future hydropower dams in the Tropical Andes, Colombia provides important lessons: First, loss of river sediments may be unavoidable, but the impact could be minimized by targeting catchments with low sediment loads or implementing sediment bypass systems where feasible. Second, downstream warm water effects are probably unavoidable unless intake systems can mix surficial and deep waters to match upstream temperature. Finally, cold, anoxic discharges may be avoidable through siting/design choices. High elevation reservoirs (e.g., Chivor, Guavio) and those in well-preserved catchments with low levels of biochemical oxygen demand (La Miel) will be less prone to developing anoxia. Short residence times may prevent reservoirs from discharging hypoxic waters downstream (Porce 3). Multiple intakes spanning a range of depths (Chivor) or may help avoid downstream hypoxia. Failure to design hydropower schemes for effective oxygen management may later necessitate costly interventions, such as liquid oxygen injection (Quimbo), or exemptions to environmental protection policies with considerable impacts (Urrá).

The energy sectors from across the Tropical Andes would be wise to take into consideration the ecological successes and failures of Colombia's existing hydropower portfolio as they decide which of the many hundreds of potential hydropower projects to prioritize. Our findings on the pervasiveness of challenges with sediment loss, thermal regime change and hypoxia could be incorporated in existing basin-scale planning frameworks that already use multiple indices to avoid and minimize anticipated environmental and social impacts of hydropower expansion (Opperman et al., 2015, 2017). The environmental impacts we document in Colombia may also be useful for evaluating specific hydropower project plans. Due

diligence during dam planning will not only help industry comply with environmental regulations, but also potentially
optimize the net value hydropower can deliver to society, by avoiding and minimizing external environmental costs.

## Code Availability

RSW wrote R scripts to visualize monitoring data .csv files extracted from the ANLA GDBs by DV. The R scripts and .csv files are available from the corresponding author (RSW) on request.

## Data Availability

Data are public and may be requested directly from ANLA, which has an open data policy https://datosabiertos-anla.hub.arcgis.com/. Alternatively, the .csv files analyzed in this manuscript may be requested directly from the corresponding author.

## Author Contributions

RSW, SLC, JD and LJS conceptualized the experiments. DVR and CBF curated the data. RSW analyzed the data. RSW, BW
and LJS secured funding. All authors participated in the research investigation. BW and LJS administered the project. RSW wrote R scripts to analyze and visualize the data. LJS and BW provided supervision. RSW wrote the original draft. All authors participated in the editing and review of the manuscript.

## Acknowledgements

This work was supported by a grant from the Leading House for the Latin American Region, Centro Latino Americano-
375 Suizo of the University of St. Gallen (CLS HSG) mandated by the Swiss State Secretariat for Education, Research and Innovation (SERI). We acknowledge additional in-kind support by the University of Antioquia and The Nature Conservancy. We thank the Autoridad Nacional de Licencias Ambientales for making their data publicly available and Silvia Vanegas Pinzón for facilitating access. Thanks to Juan Sebastian Hernandez Suarez for helpful suggestions that improved the manuscript.

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

**Table 1.** Summary of available data in samplings per year for assessing reservoir stratification (no. of profiles), changes to river temperature, dissolved oxygen and total suspended solids caused by dams for all hydropower stations licensed by the Autoridad Nacional de Licencias Ambientales (ANLA) in Colombia. Power plants with data for upstream versus downstream comparisons are listed first and sorted by surface area.

| Powerplant | Elevation (masl) | Built | Reservoir km² | km³ | Profiles n y⁻¹ | Temp. | DO | TSS |
|---|---|---|---|---|---|---|---|---|
| | | | | | | up vs. downstream pairs y⁻¹ | | |
| Quimbo | 600 | 2015 | 83.2 | 1.82 | 5 | 5 | 5 | 5 |
| Urra | 70 | 2000 | 74.0 | 1.74 | 12 | 12 | 12 | 0 |
| Sogamoso | 175 | 2015 | 70.0 | 4.80 | 0 | 2 | 2 | 4 |
| Betania | 561 | 1984 | 68.9 | 1.97 | 2 | 2 | 2 | 1 |
| Guatape | 983 | 1976 | 51.4 | 1.24 | 2 | 2 | 2 | 0 |
| Prado | 361 | 1971 | 33.8 | 0.97 | 0 | 1 | 1 | 1 |
| Calima | 1408 | 1964 | 21.0 | 0.53 | 0 | 1 | 1 | 0 |
| La Miel | 700 | 2002 | 13.6 | 0.57 | 6 | 6 | 6 | 6 |
| Chivor | 1258 | 1976 | 12.0 | 0.76 | 3 | 3 | 3 | 3 |
| Guavio | 1640 | 1989 | 13.3 | 1.04 | 0 | 3 | 3 | 3 |
| Rio Grande | 2270 | 1988 | 12.1 | 0.20 | 1 | 1 | 1 | 0 |
| Porce II | 850 | 2001 | 8.9 | 0.14 | 2 | 2 | 2 | 2 |
| Porce III | 700 | 2011 | 4.7 | 0.17 | 12 | 12 | 12 | 12 |
| Playas[*] | 983 | 1986 | 4.4 | - | 2 | 2 | 2 | 0 |
| Punchina | 775 | 1982 | 3.4 | 0.07 | 3 | 2 | 2 | 2 |
| Ituango | 300 | 2018 | 38.1 | 1.63 | 0 | 0 | 0 | 0 |
| Salvajina | 1100 | 1985 | 22.1 | 0.76 | 0 | 0 | 0 | 0 |
| San Lorenzo[*†] | 1247 | 1988 | 10.7 | - | 3 | NA | NA | 0 |
| Miraflores[*] | 2062 | 1965 | 8.0 | - | 0 | 0 | 0 | 0 |
| El Paraiso[*] | 2564 | 1950 | 6.2 | - | 0 | 0 | 0 | 0 |
| Anchicaya | 655 | 1952 | 1.4 | 0.05 | 0 | 0 | 0 | 0 |
| San Francisco | 1300 | 1969 | 0.8 | 0.01 | 0 | 0 | 0 | 0 |

[*]Reservoir volume not known

[†]Turbinated discharge goes directly into Playas Reservoir, so there is no downstream river to compare in inflowing conditions.

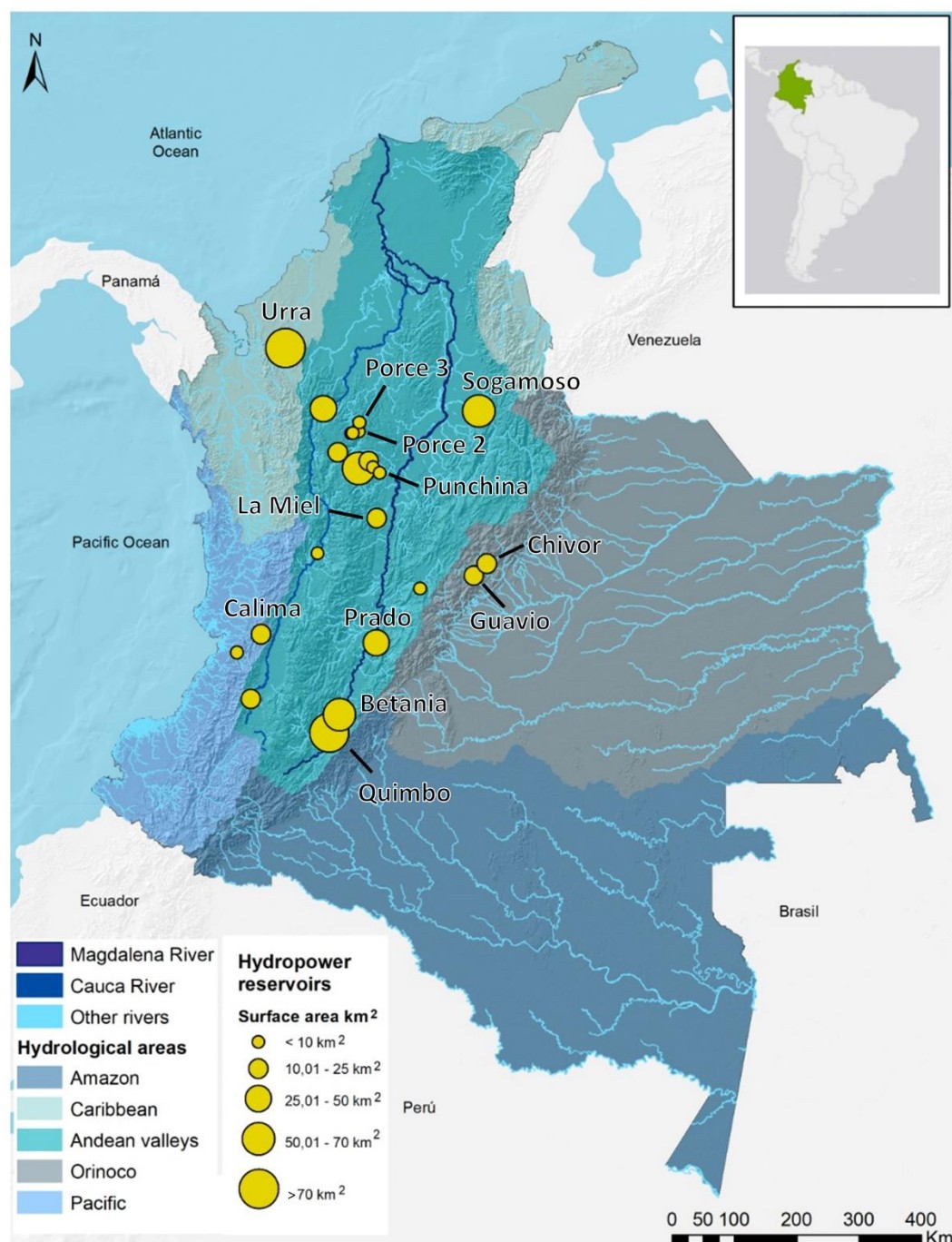

**Figure 1: Map of 22 reservoir-forming hydropower stations in Colombia regulated by the Autoridad Nacional de Licencias Ambientales. Labelled reservoirs are those analyzed in this study. Topographic base map is publicly available from the Sistema de Información Ambiental de Colombia and locations of hydropower projects are provided by Sistema de Información Geográfica maintained by Autoridad Nacional de Licencias Ambientales.**

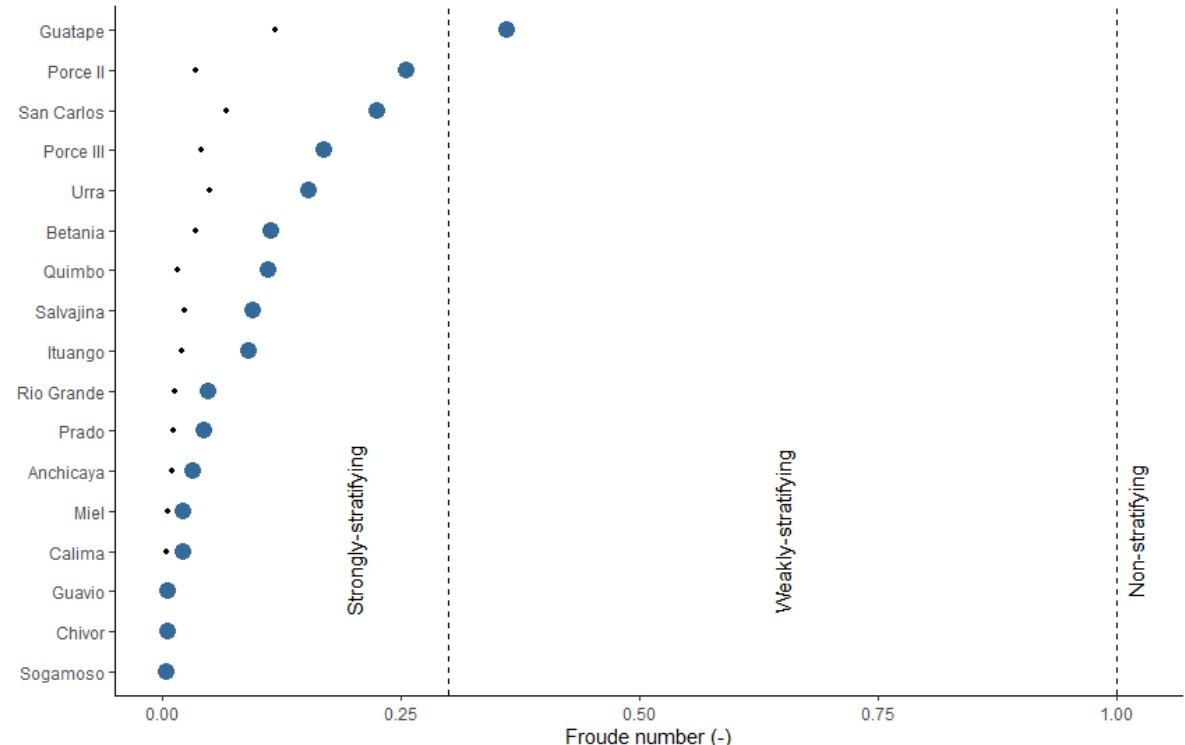

**Figure 2: Colombian reservoirs sorted by densiometric Froude number (Fr), which is a function of reservoir depth, length, volume and discharge (Parker et al., 1975). The vertical lines at Fr = 0.3 and Fr = 1.0 indicate the expected boundaries between strongly-, weakly- and non-stratifying water bodies (Orlob, 1983). Small dots represent Fr if maximum depth (height of dam wall) is used instead of mean depth, as recommended by (Ledec and Quintero, 2003). Underlying data sourced from Autoridad Nacional de Licencias Ambientales and the International Commission on Large Dams (World Register of Dams, 2018) (https://www.icold-cigb.net/, accessed: 17 November 2020). Five reservoirs from Table 1 are excluded because of missing discharge data.**

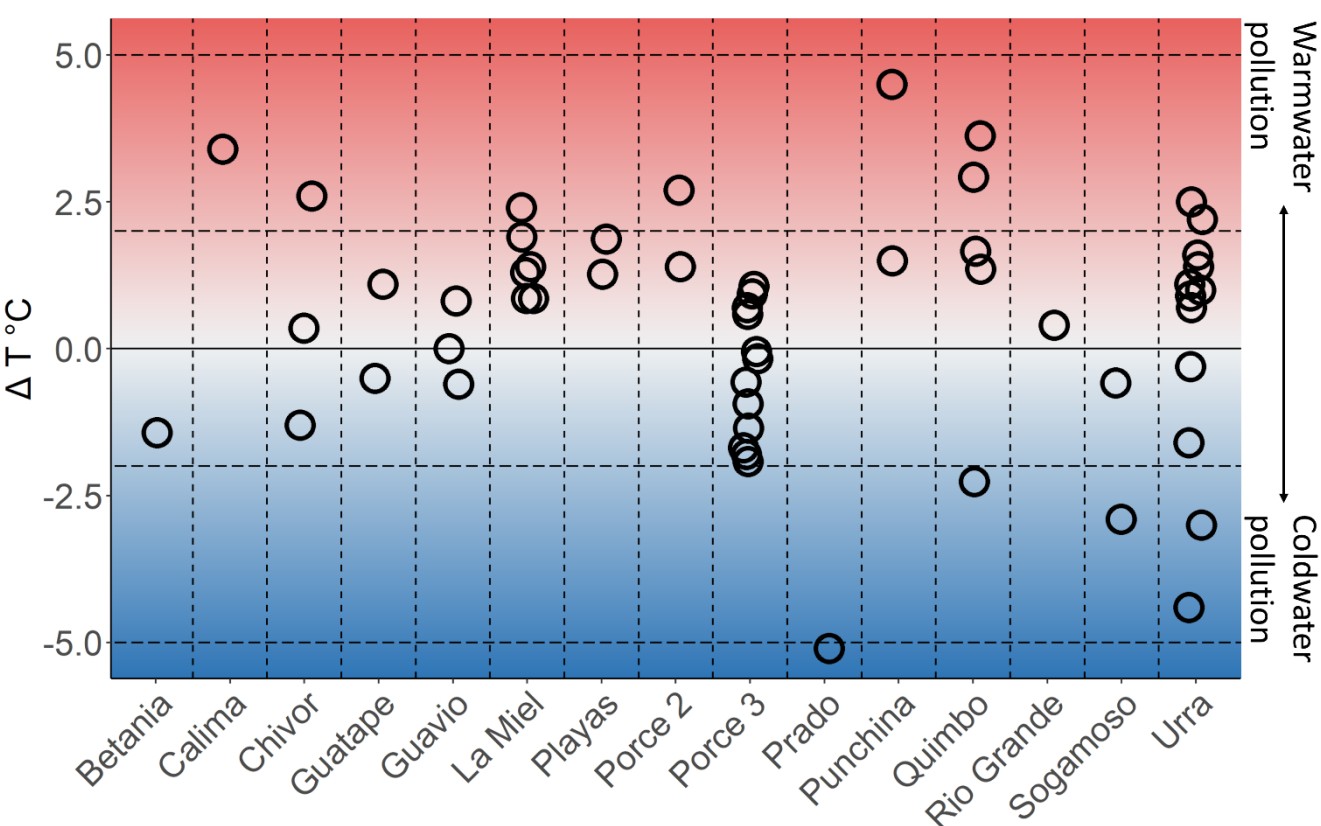

**Figure 3: Temperature differences between upstream and downstream river surfaces of 15 hydropower dams in Colombia. Each point represents one pair of contemporaneous measurements from the most recent year of available data (either 2018 or 2017).**

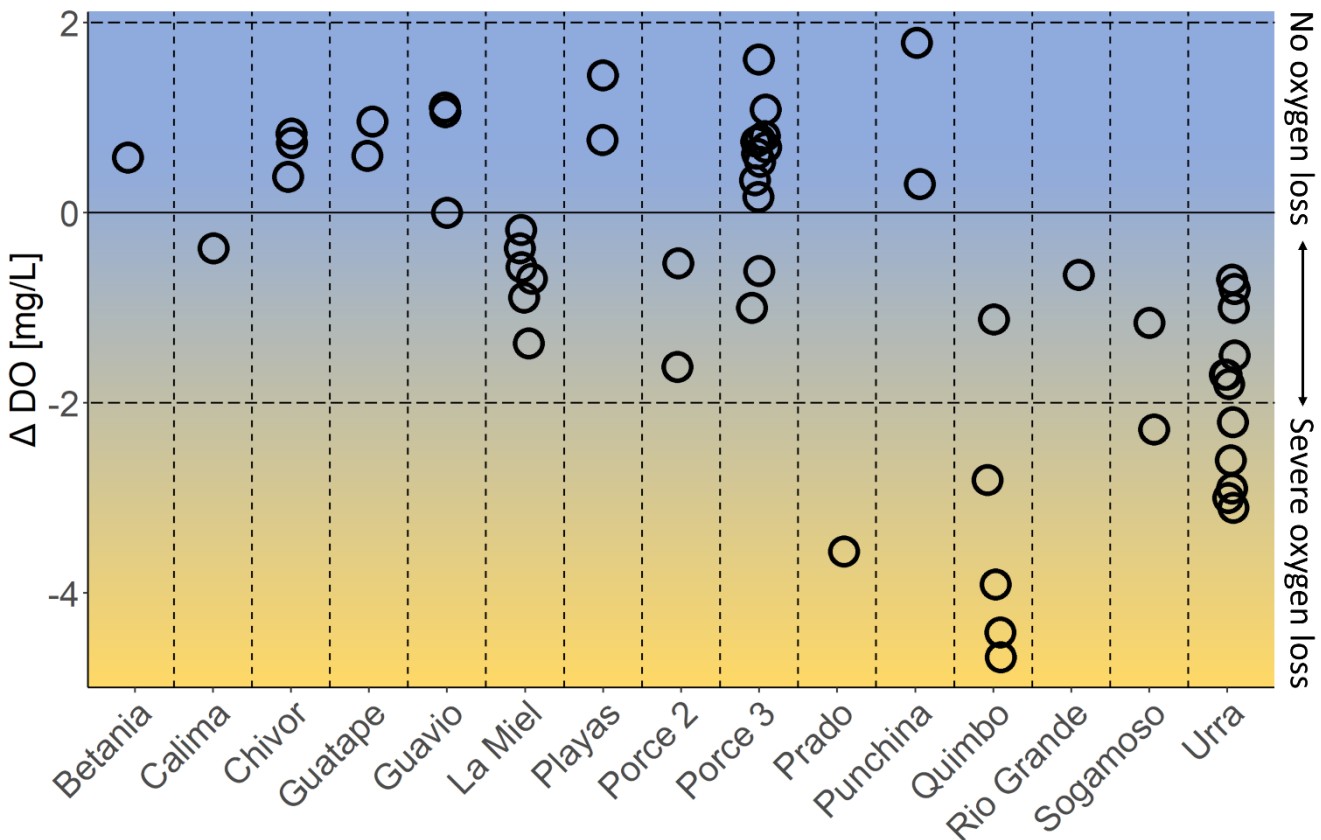

**Figure 4: Differences in dissolved oxygen between upstream inflowing river surface and river surface downstream 15 hydropower dams in Colombia. Each point represents one pair of contemporaneous measurements from the most recent year of available data (either 2018 or 2017).**

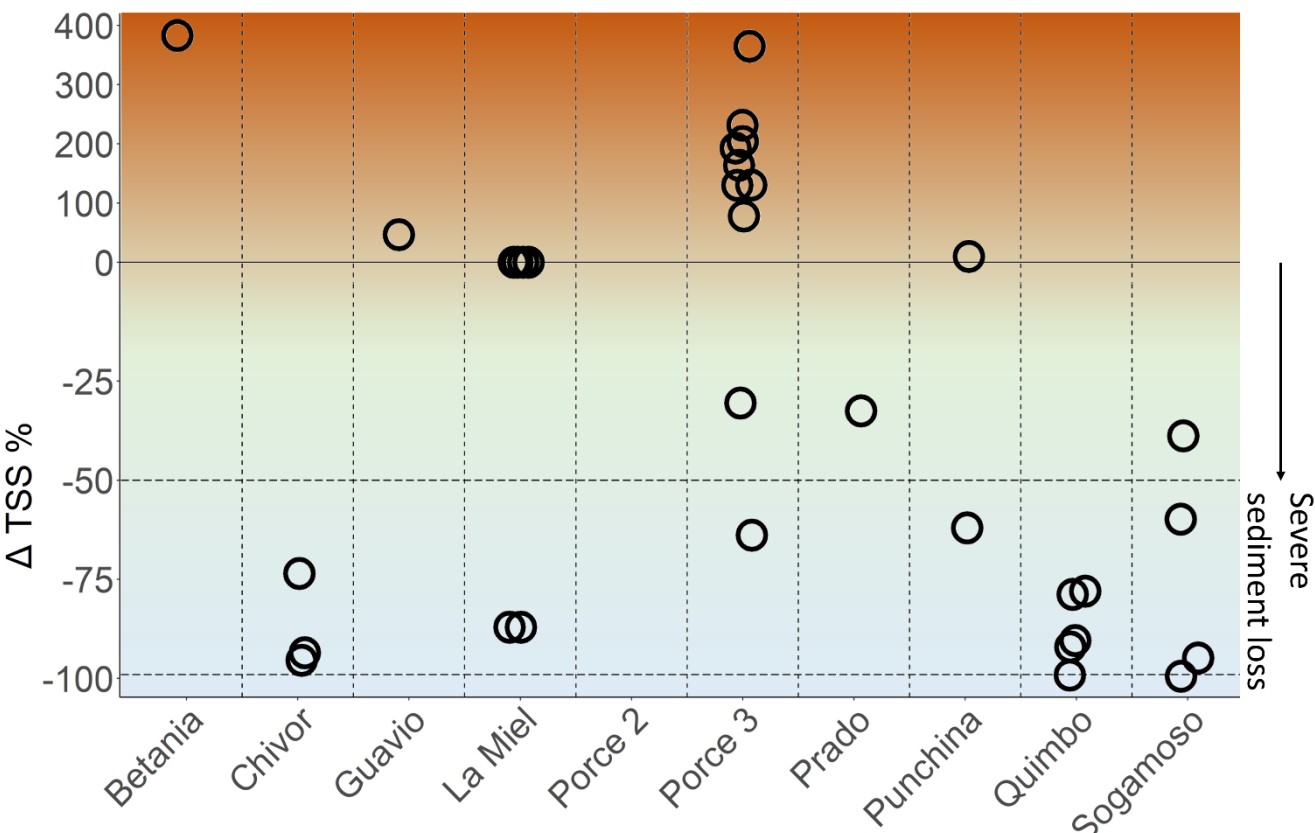

**Figure 5: Proportional change in mass of total suspended solids (TSS) in river water downstream of 10 Colombia hydropower dams relative to upstream values. Negative values indicate loss of TSS, positive values indicate increase in TSS. Each point represents one pair of contemporaneous measurements from the most recent year of available data (either 2018 or 2017).**

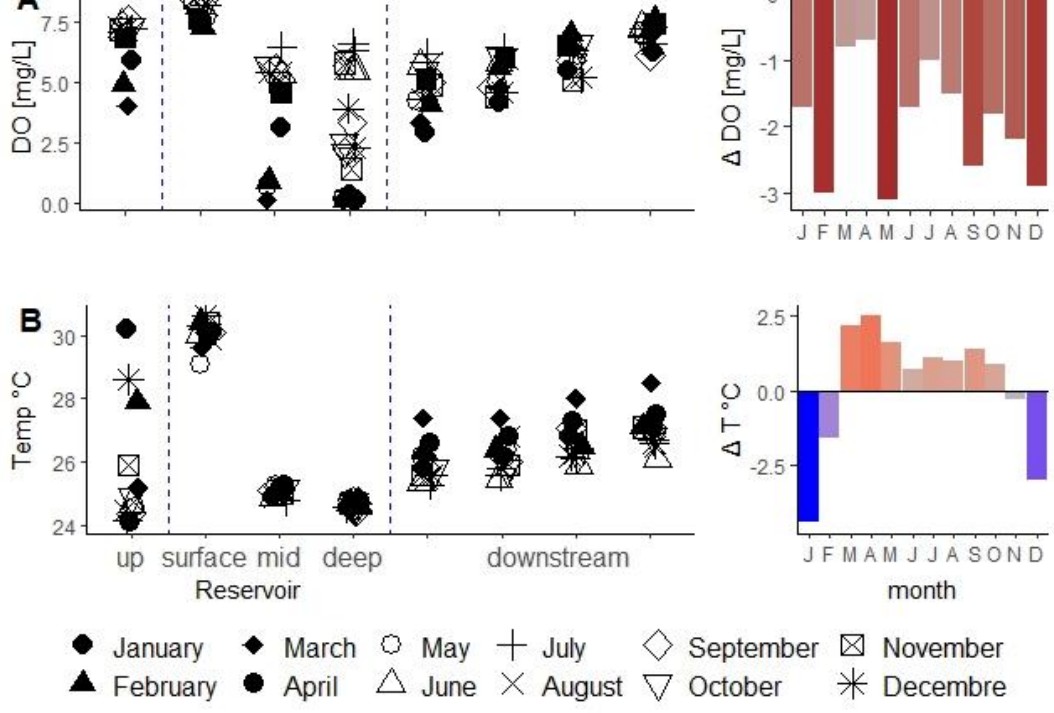

585

**Figure 6: Monthly monitoring data from the Urrá Reservoir in Cordoba, Colombia in 2018 showing upstream, reservoir, downstream as well as the upstream-downstream difference for A) Dissolved oxygen concentration and B) temperature. Data source: Autoridad Nacional de Licencias Ambientales. Data accessibility is described in Methods section.**