# Peer review of "Patterns and drivers of water quality changes associated with dams in the Tropical Andes"

_EGUsphere, 2022_

## Author Response (AR1)

Reviewer #1

The ms is extremely well written and easy to read. The approach is very well thought through, and the findings and recommendations clear. There are some minor typological corrections that can be picked up by a copy editor. I would emphasise that is it extremely rare for me to come to such a conclusion on the first review. Indeed I only remember such a positive reaction to one other ms in the hundreds of reviews that I have done.

Author Response:

We thank the reviewer for the kind comments and are pleased to know that they enjoyed our work.

Reviewer #2

Overall the paper has strong scientific significance and quality as well as strong presentation quality. The authors show how Colombian dams drive changes in temperature, dissolved oxygen, and suspended sediment in rivers using monitoring data. They also show if and how stratification is observed in reservoirs, and how that relates to downstream impacts. They connect their findings with the drivers such as the upstream catchment properties and dam design, and thus make a strong practical connection. The impacts on aquatic ecosystems of the tropical Andes are also presented. The authors end with clear and concise recommendations to regulators and planners, which makes it potentially highly impactful and socially-relevant. The authors connected all of these themes seamlessly and made it an enjoyable and insightful paper to read.

The authors were working with a limited dataset due to the data scarcity in the tropical Andes, which is a common challenge in other parts of the world where dams are booming. Thus, this paper has implications beyond the tropical Andes. The figures communicate clearly communicate important information and are helpful models for other studies to follow.

I have made minor comments which I hope will help make the manuscript even stronger, as outlined below. While the literature review is overall good, I believe the paper would benefit integrating information from other literature in other places, as I mention in the specific comments. I also believe that the authors have strong communication to regulators and planners, but could make some slight improvements for an even stronger impact. Lastly, I think that there could be a few more sentences towards the end that connect to the regional and global implications of the study (I didn't have any specific comments on this). I think it's pretty clear from the introduction, however would be interested to see more directly what the broader implications are for reservoir impacts on water quality from the perspective of the authors. Thank you for this important work!

Author Response:

We are pleased that the reviewer found this work to be of scientific significance and high quality and they recognize its relevance and potential for impact. We are also grateful for the many thoughtful suggestions for ways in which our message could be strengthened through a minor revision. All of these recommendations are highly constructive, and we plan to utilize them all to communicate this research more effectively. We summarize/categorize the recommendations briefly and provide the following responses:

-The reviewer gave several suggestions for adding important connections to key existing literature with specific articles identified that support specific paragraphs of our text.  We will look for ways to cite each of these recommended articles.

-The reviewer provides some important advice for how the work might be perceived by regulators and suggests a few changes that could help communicate regulatory implications a bit more clearly. This includes suggestion the addition of some text to the end that explains how these lessons might apply to regions beyond the tropical Andes. Since we also hope that this work will be influential for the regulatory community, we will seek to address these points carefully and provide more explicit and clear messaging.

-The reviewer suggests some modifications to the figures, which we think are all worth implementing. The labels of the dams in Fig. 1, we had already added, but by mistake uploaded an unlabeled version, a regrettable oversight that we will correct.

-The question about potential for satellite-based monitoring is an interesting one. Although remote sensing of water quality is not a specialty of our author team, we can

comment on why the Tropical Andes might be a challenging region to pursue this option. The region is extremely humid and finding cloud-free optical imagery is difficult in many regions for the wet season (when sediment mobilization is at its peak) and in some regions, such as the Choco, almost no cloud-free imagery exists. We note a recent review on this topic in S. America (Sheffield, J., Wood, E.F., Pan, M., Beck, H., Coccia, G., Serrat-Capdevila, A., Verbist, K., 2018. Satellite Remote Sensing for Water Resources Management: Potential for Supporting Sustainable Development in Data-Poor Regions. Water Resour. Res. 54, 9724–9758. https://doi.org/10.1029/2017WR022437), which is focused on hydrology, but does not mention water quality. Remote sensing could help with temperature, turbidity and chlorophyll -a, but cannot directly detect dissolved oxygen.

-The reviewer asks a specific question about data availability, which we should answer: Our goal was to look across the diversity of hydropower projects and so we requested from ANLA (the regulatory authority) the most recent year of vetted and approved monitoring data for each dam. There is historic data that ANLA is working to integrate it its modern data set, which should eventually allow the public to access real and historical monitoring information. Since we were not interested in assessing evolution of behavior over the years since construction we focused on a recent year where we could be sure to get contemporaneous data spanning as many projects as possible. ANLA is working to make all of its data publicly accessible, but while the data portal remains in development, provision of data is provided in response to goal-oriented requests, so we were provided the data that we requested based on our research goal.

-The reviewer makes a variety of recommendations to improve clarity or slightly modify phrasing to better reflect reality. All these recommendations seem very sensible and worth addressing.

-The reviewer also took the time to correct some objective errors in grammar and typos. We thank the reviewer for kindly pointing out these mistakes!

**Specific comments:**

**Lines 38-39:** I question the use of "unintended"... it seems that consequences of dam construction now are well-known globally so I'm not sure that it can be generalized that consequences are unintended.

AR: A fair point. We have changed the wording to imply merely negative side effects without implying intent (benign or otherwise).

Commented [WL1]: … but negative side effects of dam construction may create obstacles for achieving the SDGs.

Commented [ss2R1]: Do it

**Line 55:** A recent and relevant study that you may consider citing here is the following:

Flecker, Alexander S., Qinru Shi, Rafael M. Almeida, Héctor Angarita, Jonathan M. Gomes-Selman, Roosevelt García-Villacorta, Suresh A. Sethi, et al. "Reducing Adverse Impacts of Amazon Hydropower Expansion." Science 375, no. 6582 (February 18, 2022): 753–60. https://doi.org/10.1126/science.abj4017.

AR: Thanks. We will cite this below as an example of a regional study.

**Lines 55-58:** It seems that some of these are specific to the Amazon, not global studies as the sentence implies. Consider using all global studies. Some examples to be considered are below:

Fragmentation:

Grill, G., B. Lehner, M. Thieme, B. Geenen, D. Tickner, F. Antonelli, S. Babu, et al. "Mapping the World's Free-Flowing Rivers." Nature 569, no. 7755 (May 2019): 215–21. https://doi.org/10.1038/s41586-019-1111-9.

Sediment:

Vörösmarty, C. J., Meybeck, M., Fekete, B., Sharma, K., Green, P., & Syvitski, J. P. M. (2003). Anthropogenic sediment retention: Major global impact from registered river impoundments. Global and Planetary Change, 39(1–2), 169–190. https://doi.org/10.1016/S0921-8181(03)00023-7

Syvitski, J. P. M., Vorosmarty, C. J., Kettner, A. J., & Green, P. (2005). Impact of Humans on the Flux of Terrestrial Sediment to the Global Coastal Ocean. Science, 308(April), 376–381.

AR: Good point. We have replaced Anderson et al with G. Grill et al. (2015) An index based framework for assessing patterns and trends in river fragmentation and flow regulation by global dams at multiple scales. Env. Res. Lett 10 doi:10.1088/1748-9326/10/1/015001. We have also cited. Vorosmarty et al 2003.

**Line 61:** Flecker et al. (2022) (cited above) could also be a good example of a regional study. There are also several examples of strong regional studies from the Mekong river basin, such as the following:
* * *
Commented [WL3]: I would cite this as suggested on line 61 as a regional study.

Commented [ss4R3]: Do it

Commented [WL5]: Fragmentation: deleta Anderson, replace by
G. Grill et al. (2015) An index based framework for assessing patterns and trends in river fragmentation and flow regulation by global dams at multiple scales. Env. Res. Lett 10 doi:10.1088/1748-9326/10/1/015001.

GHG delete Almeida, leave Harrison which is global

water supply – Oppermann is global

water quality – Winton is global

Commented [ss6R5]: Do it

Commented [WL7]: in my view, the 2015 by Grill et al. cited above is more into fragmentation.

Commented [WL8]: Good suggestion to add sedimentation and the 2003 paper is a classic (more detailed than the sequel 2005).

Commented [ss9R8]: Do it

Commented [WL10]: Good idea to cite these two regional studies

Commented [ss11R10]: Do it

Kummu, M., X. X. Lu, J. J. Wang, and O. Varis. 2010. "Basin-wide sediment trapping efficiency of emerging reservoirs along the Mekong." Geomorphology 119 (3–4): 181–197. https://doi.org/10.1016/j .geomorph.2010.03.018.

AR: We have added these citations.

**Line 70:** These parameters "are fundamental to the condition of aquatic ecosystems" seems vague. Perhaps it should be "are fundamental to understanding the condition".... Or discussion about how the parameters being in specific ranges is fundamental to maintaining healthy aquatic ecosystems.

AR: we have reworded this sentence to be more clear/direct

**Line 99:** Consider providing the link to the data source

AR: As we explain below, there is no open access data portal (yet).

**Lines 101-102:** Were you only able to access/analyze 2017-2018 data, and not other years? Or did you only select those years? Perhaps it would help to clarify so that readers better understand accessibility.

AR: We have clarified how we accessed data and why we only requested data from 2017/2018.

**Line 124:** I think "We feel that" should be removed as it dilutes the recommendation and makes it seem like an opinion that can more easily be ignored by regulators

AR: agreed. fixed

**Lines 212-215:** This is an important point- perhaps it can be stated earlier in the paragraph as a topic sentence.

AR: good idea. We have implemented this suggestion

**Line 264:** See Dunn et al 2019 as another relevant (and more recent) study

Dunn, Frances E, Stephen E Darby, Robert J Nicholls, Sagy Cohen, Christiane Zarfl, and Balázs M Fekete. "Projections of Declining Fluvial Sediment Delivery to Major Deltas Worldwide in Response to Climate Change and Anthropogenic Stress." *Environmental Research Letters* 14, no. 8 (August 1, 2019): 084034. https://doi.org/10.1088/1748-9326/ab304e.

**Commented [WL12]:** Suggestion: These parameters are … define the critical oxic living conditions for aquatic fauna and the dynamic reconstruction of riparian habitat.

**Commented [ss13R12]:** Do it

**Commented [WL14]:** We explain below that there is no portal.

**Commented [WL15]:** I agree

**Commented [ss16R15]:** Do it

**Commented [WL17]:** Good idea to move that up to Line 200 ff

**Commented [ss18R17]:** Do it

**Commented [WL19]:** Good suggestion

**Commented [ss20R19]:** Do it

AR: good suggestion. We have added this citation.

**Lines 278-311:** This section is strong, however mainly focuses on the need to increase monitoring frequency. There were other regulatory implications discussed throughout the text, such as the need to change the temperature regulation from 5 deg C to 2 deg C. I think it would help to briefly reiterate those various items in this section. It's possible a regulator would skim the rest of the paper and look closely at this section— what would be the most important things to reiterate?

AR: We have to be careful with specific water quality indicators and specific changes to policies as this study is not a critical review or meta-analysis that would be required to support such a recommendation. We do appreciate the thought about seeing this through the regulator's eyes who may not read other sections of the paper and add an opening sentence stating the simpler more obvious idea that problems for T, DO and sediment are common and should be explicitly addressed in environmental impact statements for new dams.

**Line 302:** Would satellite remote sensing be another viable option for monitoring some of the parameters, for at least a first-order approximation? Since in situ monitoring in the Andes is challenging, it seems to be a practical option to consider. There would be several limitations to consider, of course. I am aware of studies in the Mekong River basin where satellite data is used to monitor impacts of dams on sediment and temperature—see citations below (you don't necessarily need to cite them in your paper, but perhaps they could help your investigation)

Bonnema, Matthew, Faisal Hossain, Bart Nijssen, and Gordon Holtgrieve. "Hydropower's Hidden Transformation of Rivers in the Mekong." Environmental Research Letters 15, no. 4 (April 1, 2020): 044017. https://doi.org/10.1088/1748-9326/ab763d.

Beveridge, Claire, Faisal Hossain, and Matthew Bonnema. "Estimating Impacts of Dam Development and Landscape Changes on Suspended Sediment Concentrations in the Mekong River Basin's 3S Tributaries." Journal of Hydrologic Engineering 25, no. 7 (July 2020): 05020014. https://doi.org/10.1061/(ASCE)HE.1943-5584.0001949.

AR: This is an interesting suggestion. We added a few sentences to acknowledge the possibilities/capabilities and the short-comings of remote-sensing as an approach to monitoring river conditions.

**Figure 1:** Consider labeling the hydropower reservoirs on the map, since you refer to them by name in the text/figures, and the relative location seems to be important. Or,

**Commented [WL21]:** I would be careful here. Policy propositions should be based on research and this is not a critical review or a meta-analysis that would justify specific water quality indicators.
In diplomatic terms, however, we could raise the red flags of oxygen depletion, temperature changes and sediment retention. Specific risk assessment of these negative side-effects should be included in environmental impact assessments of new dam projects.

**Commented [ss22R21]:** Make this change and explain

**Commented [WL23]:** The answer is yes for temperature and sediment turbitiy in large rivers and reservoirs. But this is very much work in progress.
Strong turbidity differences can be mapped along rivers
Rudorff, N., Rudorff, C.M., Kampel, M., Ortiz, G., 2018. Remote sensing monitoring of the
impact of a major mining wastewater disaster on the turbidity of the Doce River plume
off the eastern Brazilian coast. ISPRS J. Photogramm. Remote Sens. 145, 349–361.

**Commented [ss24R23]:** Add this text and explain

at least label the eight reservoirs in Table S3. I understand that labeling the reservoirs might made the figure too busy, but you could also use numbers to label in the map and add a table. Could also consider adding major cities, such as Medellin and Bogota (since urban effects are discussed in the text)

AR: We have added the reservoir names from table S3

**Figure 3:** Consider adding horizontal lines for + 2 degrees and – 2 degrees (they could be in another color like gray?), since that's an important threshold that you mention. Could also indicate the regulatory limit (5 deg) to emphasize the difference (which could help with messaging to regulators)

AR: We have added the horizontal lines at +/- 2 degrees

**Figure 4:** Consider adding line for -2 mg/L since this is the threshold that makes the downstream waters below the regulatory limit.

AR: we have added

**Technical corrections:**

**Line 128:** "we are unlikely to [be] capturing"—need to add "be"

AR: fixed

**Line 180:** Grammar is awkward— perhaps say "while other reservoirs" or "and other reservoirs"

AR: fixed

**Line 247:** Here you say "the authors" but in other parts you say "we" – edit for consistency

AR: We were actually referring to the authors of the study cited at the end of the previous sentence (rather than ourselves). We have changed the wording the remove this ambiguity.

**Line 258-259:** Need comma after "delivery" for consistency. Also, would be good to clarify that "downstream" refers to reaches downstream of the dam but upstream of

the delta. Consider breaking this into two sentences to clarify these things as they might be confusing for people not familiar with the concept.

AR: fixed

**Line 299:** should be "justifies" and no comma is needed before that

AR: fixed

**Figure 1:** In legend for hydropower reservoirs, it should be >70 km2 (not <70 km2).

AR: fixed

Commented [ss28]: Tag camilo

**Figure 4:** Y-axis on left side should be "DO" not "OD"

AR: fixed

Commented [ss29]: Do it

**Table S2:** "Reserervoir" should be "Reservoir"

AR: fixed

Commented [ss30]: Do it

Reviewer #3

The paper "Patterns and drivers of water quality changes associated with dams in the Tropical Andes" by Winton et al. presents an assessment of the effect of Colombian dams on downstream water quality, specifically focusing on temperature, oxygen availability and sediment loss. Even though I find the paper clearly written, I think that is has some methodological drawbacks that need to be clarified. Given the relevance of the paper for the management of water resources in the tropical Andes, I consider it could be suitable for publication in HESS after some points described below are implemented in the manuscript.

General comments

Data quality assessment: Even though the authors mention that the presented data has been curated by the ANLA (L.309), there is no information on how this procedure was carried out, so that the presented data can be reliable. Also, the authors must have

carried out a quality control of the available data before using it in their analyses. This procedure is remarkably important considering that the data in the ANLA "repository" comes from different sources. The methodology should clearly specify how such a procedure was conducted so that the use of all presented data and the results from their analysis are justified.

 AR: we have added details about the data QA/QC process carried out by ANLA and by our author team.

Measurement methods: The paper should include a table summarizing the methods used for obtaining the presented data (e.g., measurement method, accuracy) at each sampling site as supplementary material.

AR: Upon re-reviewing the original field reports from monitoring we find that the consulting companies report the method they used (e.g. SM 2550 B from Standard Methods for
the Examination of Water and Wastewater 22th edition) and show photos of temperature probes, oxygen probes, secchi disks, etc. deployed in the field, but do not actually state the device models or the manufacturer's stated precision (see screen capture of "Fig. 18" below from the monitoring report for La Miel). So unfortunately, we are not able to summarize this information in a supplemental table as requested. This is not a major problem in our view as the typical instrument precision is less relevant than the relative accuracy between different measurements with the same instrument in the field. Environmental temperature probes are typically precise to +/- ~0.01 C. Although their absolute accuracy depends on their certification and their shelf life, we emphasize the fact that we are looking for differences of 2° C as a threshold for ecologically meaningful change. Relative accuracies of commercial temperature sensors are typically two orders of magnitude better. For total suspended solids the huge changes we observed of 50 to 99% loss of TSS are two orders of magnitude beyond the precision of turbidity probes. For oxygen, the most important metric relates to the issue how reproducible DO values are following calibration and re-calibration. These metrics that are not typically reported. But again, we are interested in detecting relative differences on the order of mg / L, measured by the same person with the same probe during the same field campaign. The risk of measurement bias by reproducibility of DO – values is therefore minimized.

Commented [ss31]: Do this

**Figura 18.** Metodología de muestreo para la toma de datos de variables físicas y químicas in situ y medición de la zona fótica en el embalse.

[Figure]

**Tabla 6.** Métodos y técnicas de análisis en campo (In Situ)

| Parámetro | Unidades | Método de Ensayo | Técnica |
|---|---|---|---|
| pH | --- | SM.4500-H+ | Electrometría |
| Temperatura | °C | SM.2550 B | Electrometría |
| Oxígeno disuelto | mg O$_2$/L | S.M. 4500-O C G | Electrometría |
| Conductividad | uS/cm | S.M. 2510-B | Electrometría |

Description of study sites: Section 4.1 nicely describes some specific features of the monitoring sites which are important for interpretation. However, the paper would benefit is such features are described earlier in the manuscript (e.g., a new section 2. Description of study sites). The study areas should be described in the paper and summarized in a table including relevant information such as coordinates, elevation, state/province, depth, length and volume of reservoir, management. Part of this information is currently presented in tables and figures in the supplement, but is it

important that the reader has direct access to it in the main body of the paper. Such information could be added to Table S1 and embedded directly in the manuscript.

AR: We have added some details about the elevation, geographic and climate setting for the Colombian dam sites as a whole in the methods section. Site specific information is not needed so early in the manuscript (and would add a lot of length to the document), but is important context for assessing drivers in the discussion, so we leave this content where it is. Table S1 cannot become too crowded if it is to be added to the main body of the text, so we try to keep these extra details to a minimum for space/readability. For most larger hydropower projects this type of information is freely available on Wikipedia. We do add elevation data, which is probably more useful than naming the catchment.

Temporal component: Considering the heterogeneity of the data (1 to 12 sampling dates in 2018), it would interesting to specify when the monitoring was carried out at each of the study sites (e.g., the different symbols used in Fig. 6 for Urra Reservoir). However, using too many symbols might complicate the plots visually. I suggest that the year is divided in 4 season (e.g., January-March, April-June, July-September, October-December) and that different colors per each season are used. In that way, differences among seasons could be observed and perhaps some additional insights into the role of temporal variability on the effects of dams on water quality parameters. This could even be done in current figure 6 in which the use of 12 different symbols makes it complicated to detect monthly temporal variations.

AR: The reviewer's suggestion to incorporate temporal information into the upstream-downstream comparison figures is a logical one and they helpfully anticipate challenges to its implementation (i.e. 12 symbols is probably too many). We had considered implementing something like this for Figs. 3-5, but elected not to because we worried it would distract from the central key result, which is the magnitude of contemporaneous upstream vs. downstream measurements. Binning measurements into seasons looks like a creative solution, but it is not helpful for identifying patterns because seasons are not comparable across different parts of the Andes. The main seasonal changes are in rainfall but depending on the position of the reservoir and catchment and orientation (east slope of eastern Andes, west slope of western Andes, inter-Andean valleys, etc.) there may be a bimodal or unimodal climate regime and the difference between wet and dry seasons may be more or less dramatic, and the timing of wet vs. dry seasons may be different. Additionally, since rains are conditioned by the North-South movement of the intertropical convergence zone, the latitudinal position of a reservoirs modifies the timing of the rains between northern and southern reservoirs. The reality is not so simple such that all the problems with T and DO present themselves everywhere between January and March (for example). In Fig. 6 we

use symbols and the right panel to illustrate that seasonality is important for the physico-chemical changes we observe, for this one project with monthly sampling. But for the cross-reservoir comparisons (Fig. 3-5), we don't use symbols because we don't have the data coverage assess seasonality for the entire portfolio. In summary, the sparse intra-year data coverage and heterogeneity of climate across projects make a seasonality assessment impractical (which is why we did not do it) and therefore we do not feel that adding season information to Figs. 3-5 will provide any true insights (but might even mislead where multiple well-surveyed reservoirs are clustered in the same climate/geography) and would rather make the figure unnecessarily complicated and more difficult to understand. We think there is an opportunity to examine the role of seasonality more systematically in driving the changes we observe—there is clearly a link between seasonal stratification and downstream water quality. Rather we see this as separate analysis that would be part of a future work looking at local climate, physical limnology (stratification dynamics), reservoir operation and the resulting downstream condition.

Ground water inflows: I would appreciate if the authors could comment on the potential influence of groundwater inputs on thermal and oxygen stratification in section 4.1.

 AR: We have no information about the role of groundwater inflows for any of the reservoirs and commenting on the possibilities would be pure speculation.

Specific comments:

Titile: I think the title is a bit too broad both in terms of the topic and the region. I suggest the following title that I consider more appropriate based on the presented data and geographic scope or something along those lines: "Changes in thermal regimes, oxygen availability and sediment loss associated with dams in the Colombia"

 AR: Listing the three parameters is too specific for a title. "Water quality" can never include all aspects of water quality. So the fact that it doesn't in this case as problematic for an article title. We have to mention "patterns and drivers".  It is true that we do not cover all of the tropical Andes, but as we point out in the data availability part of the methods, Colombia is the only tropical Andean country with a centralized monitoring data that allows this type of synthesis. It's dams also span a huge range of temperature and precipitation space, so it represents the region well. We worry that replacing "Tropical Andes" with "Colombia" will unnecessarily limit the paper's reach as we would really like it to get the attention of scientists working in Ecuador, Peru and Bolivia.

P1.L20: specify the number of sites included in the analysis

 AR: Fixed

P1.L22: specify percentage of all sites or number of sites

 AR: added 7 of 10

P1.L25: specify percentage of all sites or number of sites

 AR: added 8 of 15

P3.L75: not sure what this means "with hypoxia is intolerable"

 AR: reworded for clarity

P3.L92: there is only one Andean Cordillera, but with different branches. Please correct accordingly

 AR: fixed

P3.L95: specify if the companies are private, public, and/or both types.

 AR: added; they may be public or private

P4.L103: I suggest including Table S1 directly in the manuscript instead of as a supplement.

 AR: fixed

> Commented [ss32]: Do it

P4.L116: specify what "deep waters" mean in this context

 AR: fixed. depth for which minimum values are found; varies by reservoir

P4.L117: explain how the sites are assigned to these categories

 AR: added these sentences explaining more detail. "We classified reservoirs as strongly-stratifying if they showed O2 concentrations of <2 mg L-1 at depth and a difference between the surface of at least 3 mg O2 L-1. All other reservoirs we classified as weakly-stratifying as all showed differences in O2 and temperature between surface and deep waters of at least 2°C and 1 mg O2 L-1 (for summary, see Table S2)."

P4.L117-118: specify the data, equation and assumptions used to estimate Froude number

 AR: we have added much detail here about assumptions, logic and the equation for calculations

P4.L124: please rephrase without using the verb "feel"

 AR: fixed

P5.L143: here it says that data are from 2018, but in figures 2-5 it is reported (either 2017 or 2018). Please revise and correct where needed

 AR: fixed

P5.L149: Table S2 is very relevant. I suggest including it in the main body of the paper instead of as supplementary material

> **Commented [ss33]:** Do this

 AR: Yes these data are relevant, but largely duplicated by the plotted data in figures 3 and 4. Therefore we feel it is best left in the supplement so as not to crowd the main text with too many data tables that will not be likely to be of interest to most readers.

P5.L153: I suggest including Figure S1 in the manuscript and keeping S2 and S3 as supplementary material.

 AR: We do not see an improvement by adding depth profiles of T and oxygen for just one site to the main text. Much of the same information (plus a lot more) can be gleaned from Fig. 6

P6.L171: even though the text mentions 12, the figure actually shows 15. Please revise and correct accordingly

 AR: fixed

Section 4.1. The description of the sites should be included earlier in the manuscript, and here only the relevant aspects should be emphasized.

 AR: We think making this change would actually be counterproductive to manuscript readability and clarity. The site details in this section are relevant to the discussion of the drivers and they should not be separated from this discussion. There are too many sites to efficiently describe them all in the methods as site descriptions and there apart from the driver much of the information would appear irrelevant to the study setup.

P5.L190: how was the residence time of deep water estimated? Please specify

 AR: Clarified that here we are merely referring to mean hydraulic residence time rather than residence time specific to the deep water. Mean residence time is simply a function of volume and discharge, which is textbook knowledge that shouldn't require further explanation.

P5.L191: Again, I think it is worth it including Fig. S1 in the paper

**Commented [ss34]:** Do it

 AR: This would be easy to implement, but we don't see a compelling need to include this figure, which is a just a visualization of one reservoir's depth profiles, as a main text figure.

P10.L300-304: This paper (Pesántez et al., 2021) presents an automated sensor which allows measuring chemical water quality parameters in tropical Andean stream at high-temporal frequency. I think it could be useful in this part of the discussion.

 AR: Interesting paper. Added this citation with a note about possibilities for sensing some solutes.

Technical corrections:

P1.L22-25: Very long sentence, please split into 2 or more shorther ones.

 AR: fixed

P1.L30: "These findings HIGHLIGHT the importance of IMPLEMENTING environmental monitoring….."

 AR: fixed

P2.L36: "… hydropower GENERATION, which is…"

 AR: fixed

P2.L39: replace trade-off by effects

 AR: fixed

P2.L42: replace too often by frequently

 AR: fixed

P2.L56-58: use comma instead of semicolon when listing

 AR: fixed

P3.L79: replace though by through.  "… largely GOVERNS …"

 AR: fixed

P3.L82: Does stratification lead to … (i.e., delete "the" from the question)

 AR: fixed

P4.L113: qualitatively ASSESSED

 AR: fixed

P8.L230: BOCACHICO

 AR: fixed

FIGURES

Figure 1: I suggest replacing "GENERAL LOCATION" by SOUTH AMERICA in the inset map. Also, the legend of the largest circle should be >70km2, not "<"

 AR: We have opted simply delete "General location." Readers should be able to recognize the continent of South America. Fixed the  < > error.

Figure 3: add dashed horizontal lines at +2C and -2C for reference of the thresholds described in the paper

 AR: added

**Commented [ss35]:** Do it

Figure 4: add a dashed horizontal line at -2 for reference of the threshold described in the paper

 AR: added

Figure 5: add dashed horizontal lines at -50 and -99% for reference of the thresholds described in the paper

Commented [ss36]: Do it

 AR: added

SUPPLEMENTARY MATERIAL

Table S1: missing important information such as coordinates, elevation, period of sampling. Include this in the main body of the text.

Commented [ss37]: Do it

 AR: We cannot do both. In the main text the table has to only include the most relevant information. In the supplement where aesthetics are not so critical there is room for more detail. We opted to move the table as is to main text and replaced catchment name with elevation as that seems like more useful information to include.

Table S3: Specify what the colors mean

Commented [ss38]: Do it

 AR: We have clarified the rationale for highlighting the cells in the caption.

Figures S1, S2, and S3: the text of these figures are not in in English, please revise and correct.

Commented [ss39]: Do it

 AR: Fixed

REFERENCES I USED

Pesántez, J., Birkel, C., Mosquera, G. M., Peña, P., Arizaga-Idrovo, V., Mora, E., McDowell, W. H. and Crespo, P.: High†frequency multi†solute calibration using an in situ UV †visible sensor, Hydrol. Process., (August), 1–15, doi:10.1002/hyp.14357, 2021.